# Hybrid Joining by Induction Heating of Basalt Fiber Reinforced Thermoplastic Laminates

**Amar Al-Obaidi** \* , **Jonas Kimme** \* **and Verena Kräusel**

Institute of Machine Tools and Production Processes IWP, Technische Universität Chemnitz, 09126 Chemnitz, Germany; verena.kraeusel@mb.tu-chemnitz.de
\* Correspondence: amar.al-obaidi@mb.tu-chemnitz.de (A.A.-O.); jonas.kimme@mb.tu-chemnitz.de (J.K.)

**Abstract:** Induction heating was used to join basalt fiber reinforced polymer laminates (BFRPL) using the process called inductive contact joining (ICJ). Two other mechanical joining processes, nut and bolt (NB) and two-piece hollow riveting (2PR), were compared to ICJ. The obtained joints were evaluated using tensile shear tests and by analyzing fractured surfaces. Furthermore, simulation of the ICJ process was used to estimate the effective parameters. Joints produced by ICJ had superior joint strength compared to joints manufactured by 2PR. In addition, during ICJ, the BFRPL fibers were not damaged and the strength of the base material was maintained. The tensile shear forces of the ICJ process exceeded 3.5 kN and 2.5 kN for a joined, sandblasted aluminum sheet with BFRPL and for joining BFRPL to itself, respectively. Further optimization potential of the ICJ process was discovered during the investigation, so that potentially higher joint strengths and shorter processing times can be expected, making the process interesting for future industrial applications.

**Keywords:** induction heating; hybrid joining; basalt fiber; thermoplastics; FE simulation

## 1. Introduction

Fiber-reinforced polymers (FRP) are preferably used in manufacturing of lightweight structures due to their high strength, light weight, and perfect corrosion resistance. FRP are used in industrial applications, such as in the following industries: aerospace, automotive, sporting and consumer goods, machine construction, marine applications, and medical, and civil engineering. Generally, there are two types of FRP, thermoplastic and thermosetting, which depend on the polymer used in the matrix composite structure. Glass or carbon fiber reinforced polymers (CFRP) exhibit low ductility with high brittleness. A low-cost fiber, such as basalt, and the use of a thermoplastic matrix opens up new possibilities for the reproducible production of hybrid FRP. In combination with light metal cover layers, the material can be given ductility and damage resistance [1]. Therefore, implementing FRP requires a ductile metal-like steel or aluminum sheet. The joined ductile sheet metal with the FRP works as a support to prevent deformation, especially in some areas that require fixing and clamping with other parts. Many attempts were made to join FRP with metal.

Most applications of composite materials require specific fixtures to support or clamp to another metal to be fixed in the designed structures [2]. Mechanical joining methods are needed to join the polymer composites with metal sheets, like joining by extruded pins, as introduced by the authors of [3]. Another process used to join different materials depends on the plastic deformation of the workpiece, as investigated in [4]. Another joining method depends on diffusion bonding, brazing, and transient liquid phase joining processes [5] and requires high energy through pressure or heat to join metals to ceramics.

In the middle of the 1980s, induction heating was applied to join different types of polymer composites with metals [6]. Continuous movement or discontinuous tool movement was needed to press the bonding zone after heating. Induction heating was applied by the authors of [7], where two types of aluminum alloys, $AlMg_3$, and $AlMg_{0.4}Si_{1.2}$, were selected to be joined to carbon fiber reinforced nylon 66 (AP66) that was heated by

induction. Three steps were required to join the listed materials: first, heating by induction coil; second, pressing with an external tool; and finally, cooling the combination. The overall process took more than 125 s. Following the same three steps, induction heating was applied to continuous joining of titanium with CFRP, as done in [8].

A continuous seam joining was achieved using inductive heating [9]. The applied composite was a carbon-fiber fabric reinforced polyphenylene sulfide strip joined with a panel made of the same composite. The joint was made by induction heating and then pressing with a 50 mm diameter continuous roll, after which cooling by air was adopted. The processing speed was from 1 to 50 mm/s. Moreover, the joining of carbon fiber composites by direct induction heating was investigated by the authors of [10]: 60 s was required to heat the composites to 400 °C.

Another method used ultrasonic welding, as improved by the authors of [11]. Aluminum AA5754 was welded with CFRP polymer by applying a complicated and expensive system that depended on ultrasonic power energy. Additional surface treatment was required before joining and this added to the process time, but the paper did not mention how long it took to accomplish the joining.

An example of the industrial applications in automobile production is a B-pillar made of composite material that needs to be joined with other automobile body steel or aluminum structures. For this purpose, mechanical joints and chemical and thermal methods were used [7]. The applied processes were presented by clinching, riveting, gluing, and ultrasonic welding. Many disadvantages appeared with the listed joining methods, such as low strength in the joined area, long process time, high costs, and low flexibility. Furthermore, additional pre-preparations are required, like drilling holes, polishing, and surface treatment, which lead to long process times and additional costs [12–14].

Finite element (FE) simulations of induction heating thermoplastic composites have been investigated by many researchers. The LS-DYNA R7 program was used by the authors of [15] to simulate joining two layers of CFRP. After heating, a compressive pressure was exerted by a movable roll with a speed between 100–300 mm/s. Two models were investigated: with and without air cooling. They found that the temperature at the joining zone remained at a similar level and the desired bonding performance was achieved without upper surface cooling. The development of analytical models for heating-temperature evolution during the induction heating of polymer components was studied by the authors of [16]. Low-frequency induction heating was used to join plastics with a susceptor applied in the plastic packaging; the simulation was carried out using COMSOL (version 5.4a, COMSOL Multiphysics Inc, Burlington, MA, USA). They concluded that the homogeneity of heat dissipation follows a similar design to the geometry of the susceptor for the induction coil. Most of the induction simulation methods used for the heat generation calculations through a magnetic field are reviewed in [17], without taking into consideration the induction effect on workpiece mechanical properties. Finally, simulations to heat carbon fiber reinforced thermoplastic by induction have been widely investigated. An induction coil was assisted by a magnetic field concentrator in [18]. A CRFP lap joint was simulated up to a 300 °C polymer surface temperature for 5 s with uniform heat distribution. They found the optimum induction heating process by determining the combination of induction coil geometry, frequency, and power.

In this research, a new induction joining method was introduced to combine a basalt fiber reinforced polymer laminate using induction heating, compression, then cooling in the same place, with a single tool. The joined partners presented by induction heating were compared with two mechanical joining processes.

## 2. Materials Used in the Experiments

Three basic material combinations were applied in the experiments. Laminated fiber-reinforced polymer was the basic material investigated. Basalt fibers of organic natural material were combined with thermoset polymer polyamide and laminated with thin EN AW-5754 aluminum sheets of 0.3 mm thickness. Furthermore, the polymer used as

the matrix was polyamide PA6, with a melting temperature from 220 to 240 °C. The preparation and manufacturing of the laminates used in the experiments were discussed extensively by Karapepas in [1]. Basalt fibers were laminated in two samples. In the first sample, called Type I, the basalt fibers with matrix polymer were sandwiched between two aluminum sheets. The second sample, called Type II, consisted of one aluminum sheet sandwiched between the basalt reinforced polymer layers. Figure 1 illustrates the two investigated basalt fiber reinforced polymer laminates (BFRPL). The third metal used in the experiments was a 2.4 mm thick EN AW-5754 aluminum alloy sheet joined with the basalt fiber reinforced polymer (BFRP).

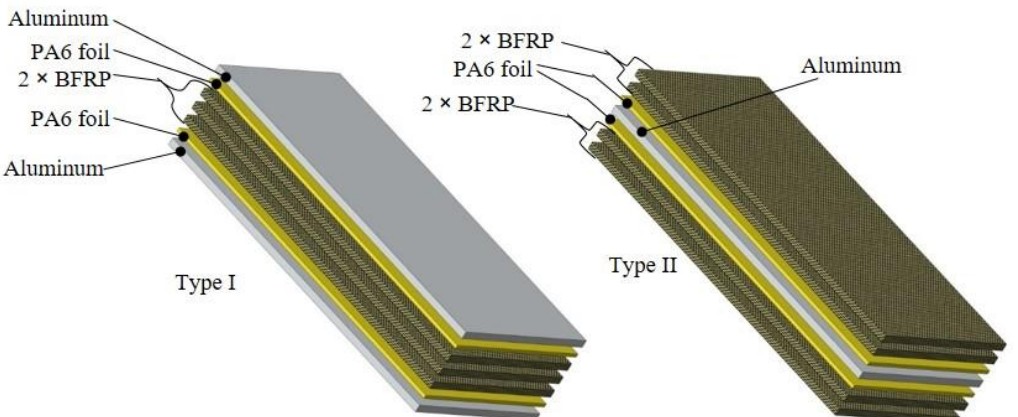

**Figure 1.** Type I and Type II basalt fiber reinforced polymer laminates (BFRPL). BFRP: basalt fiber reinforced polymer.

## 3. Joining Methods and Conditions

Three lap joining methods were investigated. The first was joining with induction heating and the other two methods were mechanical joining technologies. The results were compared with those of the induction joining process.

### 3.1. Joining by Induction Heating

This method used inductive contact joining (ICJ) using an induction coil as the heating, pressing, and cooling device all at once. As investigated by the authors of [19], the main elements affecting induction joining are the induction frequency and power generator, which are supported by current and voltage. Other elements affecting induction joining are, in order of effect, the induction coil that provides the heating source and produces the magnetic field, the workpiece, and the cooling process required to cool the joined materials. Due to the fact that both the polyamide matrix and the basalt fibers are both nonmagnetic materials, they will not be affected by induction heating. Therefore, the heat is initiated by the magnetic field that heats up the aluminum sheet and melts the polyamide matrix. The joined part is then cooled by water flow through the induction coil that is connected to a cooling system that cools the water to between 20 to 15 °C with pressure between 3 to 4 bar. Here, the polymer substrates are reconsolidated, and the joint is consolidated. Moreover, the heat and consolidation pressure are provided by the induction coil, which is made of pure copper metal. The magnetic field induces eddy currents, raising the temperature in the bonding zone. A fully detailed explanation of the ICJ process was presented by Kräusel et al. [20]. In Figure 2, the ICJ process principle is shown for joining the type II BFRPL and aluminum sheet together.

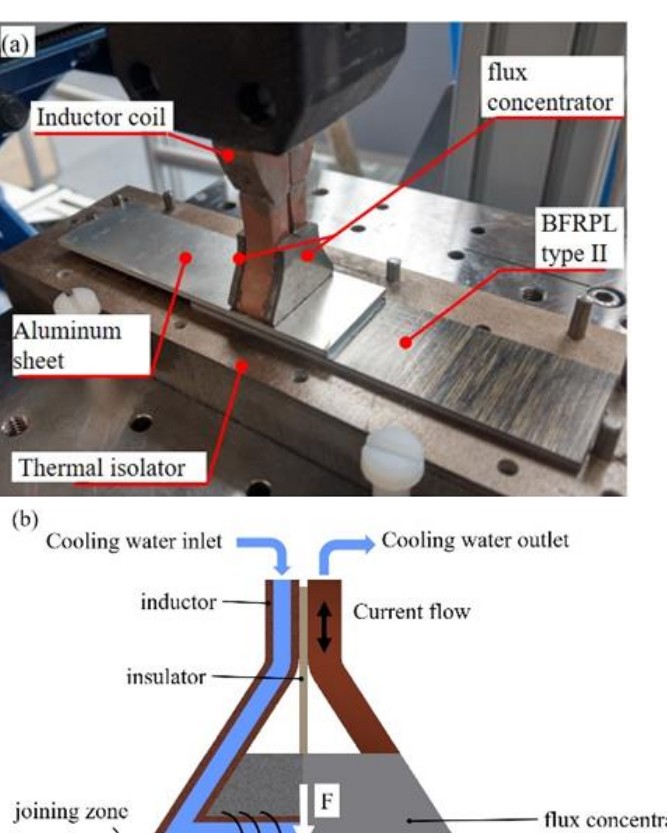

**Figure 2.** (**a**) The inductive contact joining (ICJ) experiment set-up and (**b**) schematic diagram representing the mechanism of ICJ [20].

### 3.2. Mechanical Joining Processes

A two-piece hollow riveting (2PR) joining process was used. In addition, nut and bolt joining (NB) was chosen as the second mechanical joining process. The samples were first drilled and then combined mechanically. The 2PR samples were pressed by a mechanical manual press. The NB samples were adjusted by M4 screws and nuts, then tightened together by a torque wrench with 3.5 Nm. The drilling process of the BFRPL was done using a special high speed steel (HSS) drilling tool used to drill laminate composites of both types I and II. In addition, the drilling parameters were a 0.1 mm/r feed rate and a 1000 RPM tool rotation speed.

### 3.3. Joining Samples and Process Validation

The validation of the investigated joining process was analyzed by testing the joints using a tensile shear test for spot and seam welding standards according to DIN EN ISO 14273:2014 [21]. The tensile shear tests were performed on a universal tensile testing machine at a 0.1 mm/s testing speed. The bonding zone sample dimensions were $45 \times 30$ mm; the force–displacement curves were calculated from those dimensions. The joining sample dimensions are shown in Figure 3.

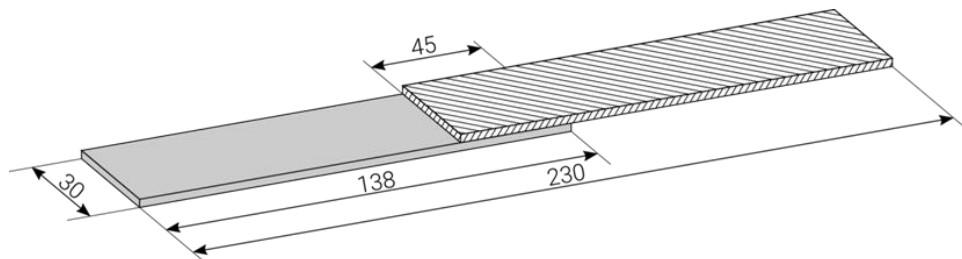

**Figure 3.** Joining sample dimensions in mm.

*3.4. Joining Sample Combinations*

Four different joining sample materials were used: aluminum (Al), sand-blasted aluminum (SAl), BFRPL Type I, and BFRPL Type II. Moreover, joining sample materials had a thickness of 2.4 mm and were cut into 30 × 138 mm dimensions by a water jet cutting machine. The sample classifications and materials combinations applied can be seen in Table 1. For each joining method, at least four samples with the different materials were made for the tensile shear tests to prove the repeatability of the investigated joining processes. All the samples for the three joining methods were investigated after joining by cutting the joined zone in two halves using a water jet machine. This made it possible to examine the joined zone with a light microscope for areas of insufficient bonding or in the case of the ICJ the existence of pores and cavities. The joined samples of Type I with Type II can be seen clearly in Figure 4.

**Table 1.** Material type and joining method applied in the experiments.

| Joining Method | Induction (ICJ) | Two Parts Hollow Riveting (2PR) | Nut and Bolt Joining (NB) |
|---|---|---|---|
| Sample material type | I+II<br>II+II<br>II+SAl<br>II+Al | I+II<br>II+II<br>I+I<br>II+Al | I+II<br>II+II<br>I+I<br>II+Al |

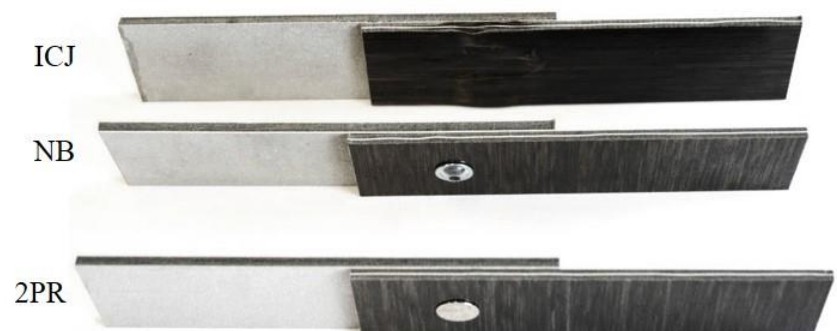

**Figure 4.** Joining samples of Type I and Type II for the three joining methods. NB: nut and bolt; 2PR: two-piece hollow riveting.

## 4. ICJ Process

The main objective of the experiments presented in this paper was to evaluate inductive contact joining to combine a composite material that is difficult to join to aluminum with an aluminum sheet. According to the literature, there have been no articles or experiments previously applied to join BFRP. Therefore, to ensure the joinability of the BFRP by the ICJ process, initial experiments were necessary. Additionally, FE simulations were required to optimize the process parameters of the experiments.

### 4.1. Initial ICJ Experiments

To join the required specified samples using the ICJ process, initial experiments were carried out to ensure the joining suitability of ICJ. Furthermore, joining parameters were determined as follows from the initial experiments that directly affect the bonded zone:

1. Heating time, where the induction power flows through the material combination,
2. Induction frequency, chosen from the high frequency and medium frequency [22],
3. Contact pressure, adjusted by a pneumatic cylinder between 2 to 4 bar, and
4. Power, required to generate an induction heat as delivered by the induction generator.

Table 2 illustrates the final effective parameters required to successfully join Type I and II BFRPL samples. In addition, experiments were carried out to heat only the BFRP without aluminum by induction heating, but the basalt fibers were not affected by induction heating. Since neither the plastic matrix nor the basalt fibers conduct the electric current, the magnetic field did not affect heating of the BFRP. Moreover, the bonding zone temperature was measured by thermocouple type K for all the three ICJ sample types.

**Table 2.** The effective parameters of the ICJ. SAl: sand-blasted aluminum.

| Sample Type | Heating Time (s) | Frequency (kHz) | Power (kW) |
|---|---|---|---|
| II+II | 4 | 350 | 6 |
| II+Al and II+SAl | 11 | 14.7 | 3 |
| I+II | 3 | 14 | 4.5 |

### 4.2. FE Simulation of the ICJ Process

For validating the joining process, FE simulation was done using COMSOL Multiphysics version 5.4a (COMSOL Multiphysics Inc, Burlington, MA, USA) of license number 6081724. A structured hexahedral mesh element was used for the joining partners (element edge length 0.04–3 mm). The remaining geometry was meshed with a tetrahedral mesh type (element edge length 0.3–40 mm). For the calculations, the AC/DC electromagnetics Module of COMSOL Multiphysics with a frequency-transient study was used. The coil current was 1150 A for type II+Al samples and 1410 A for the rest of the samples. The selected current values were determined using a Rogowski coil [23]. Heat exchange with the environment for the samples during the simulation was modeled as convective heat flux between the coil and the upper joining partner. Moreover, the heat transfer coefficient was 300 W/m$^2$·K and the coil temperature was 20 °C, elsewhere was thermally insulated. Figure 5 shows the described simulation model of the ICJ process for the type II+Al sample. The aluminum sheet was selected to be on the upper side, as in real experiments.

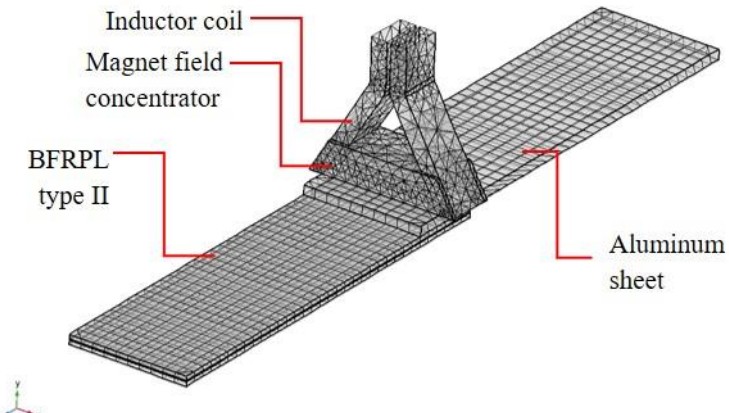

**Figure 5.** COMSOL model description showing the simulation assembly.

## 5. Results and Discussion

The results of this research include the induction heating simulation of the joined partners. Furthermore, microscopic and tensile shear tests were carried out for the three joining processes.

### 5.1. FE Simulation

The simulation results obtained using COMSOL Multiphysics provide information about the heat distribution in the ICJ process. The inductor coil temperature was assumed to be constant due to the water flow through the inductor. Therefore, during the ICJ process, the strength of the inductor coil remained unchanged. The magnetic field can only heat the metallic part of the samples because the composite material used cannot conduct an electric current due to its low permeability. Furthermore, the thermal conductivity coefficient of the BFRP is 0.0031–0.0038 (W/m·K), as observed in [24]. The aluminum sheet in the BFRL work as the added susceptors, as simulated by the authors of [16], because they are essential to magnify and heat the polymer. Whereas, in the induction heating simulation of CFRP [17], the carbon fibers can be easily affected by the eddy currents to heat the overall composite structure.

The simulated heating temperature of the ICJ process is presented in Figure 6 for the II+Al sample. Furthermore, the maximum heating temperature occurred in the aluminum sheet because it was affected by the magnetic field. The medium frequency provided by the induction generator was 10.5 s, and it was the actual heating time. This leads to melting the upper matrix layer polymer on the Type II laminate. Furthermore, after heating, the joining zone cooling effect will take place from the inductor, causing bonding between the Type II laminate and the aluminum sheet.

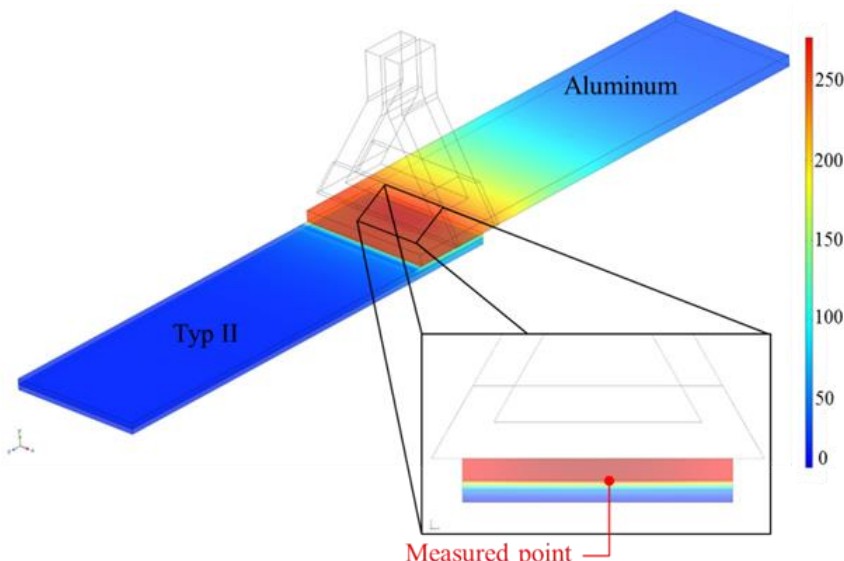

**Figure 6.** Simulation of the heating temperature of the ICJ process sample II+Al.

As illustrated in Figure 6, the simulated joining zone was cut in halves to better visualize the thickness temperatures. The highest temperature reached was 280 °C at 10.5 s, as shown in Figure 7, which is above the melting point of the PA6 applied in the type II laminate.

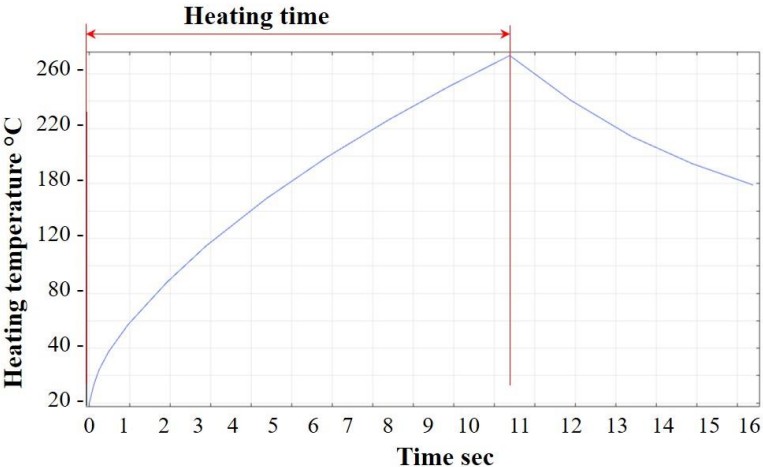

**Figure 7.** Simulated heating temperature of the II+Al sample.

The heating temperature simulation of joining type II+II is demonstrated in Figure 8. The heating time was only 4 s to heat the aluminum sheet to 510 °C using a high frequency. In addition, the high-speed heat dissipation from the thin aluminum sheet due to the heat transfer effect led to melting of the PA6. The measured heating point in the middle of the bonding section reached 230 °C. It is worth noting that medium frequency also succeeded in joining the II+II sample, but the heating time was more than 25 s.

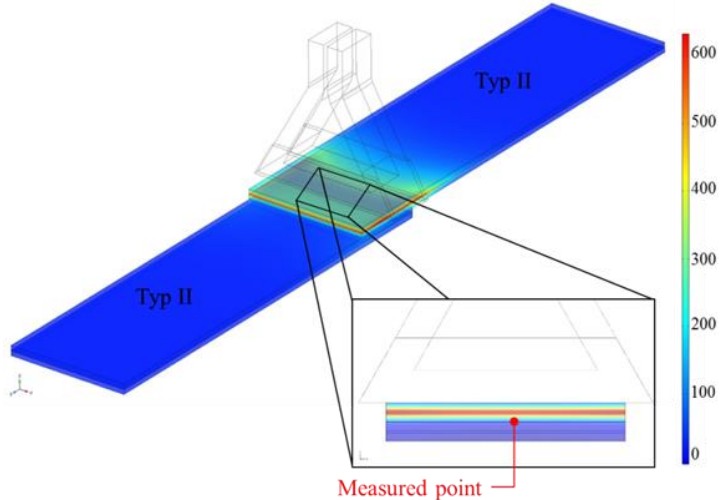

**Figure 8.** Simulation of the heating temperature of the ICJ process sample II+II.

On one hand, the medium frequency magnetic field penetrated the 2.4 mm aluminum sheet to heat the inside layer. On the other hand, the high-frequency magnetic field can only penetrate sheet metal of limited thickness. Consequently, the skin effect is directly related to the induction frequency and material permeability [22]. Therefore, medium frequency was selected for the II+Al and I+II samples.

### 5.2. Microscopic Analysis of the Joined Specimens

The joined samples were cut by water jet machine and cross-sectional view sample images were made. Figure 9 shows the cross-sectional views taken by a microscope for joining type I+II. The ICJ process mainly depends on the consolidation of the polymer matrix, as reported by the authors of [10]. As a result, the PA6 of Type II was reconsolidated over the aluminum sheet of Type I; thus, insufficient heating temperature and time led to bonding failure, as can be seen from the air gaps in Figure 9a. Consequently, inadequate consolidation pressure resulted in a deviation between the joined partners. In contrast,

successful ICJ is demonstrated in Figure 9b, showing full contact between the polymer matrix and aluminum in the bonding zone. Furthermore, mechanical joining methods of NB and 2PR processes are presented in Figure 9c,d, respectively.

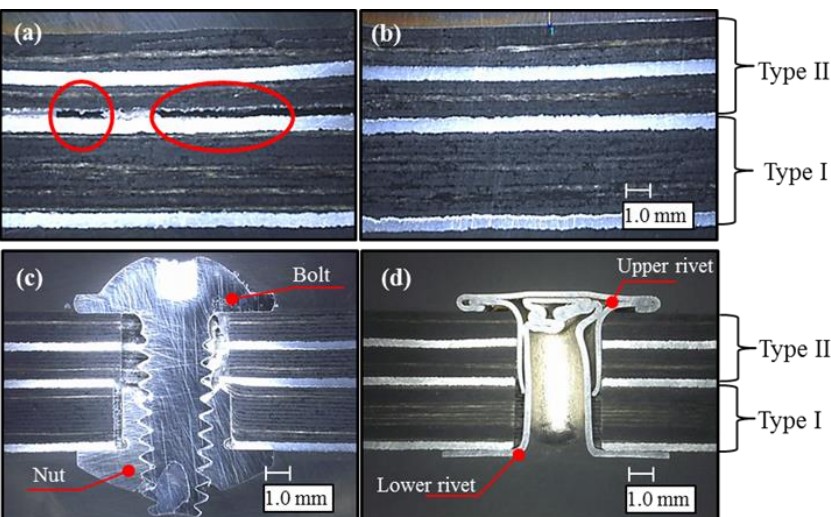

**Figure 9.** Type I+II cross sectional views of ICJ samples that (**a**) failed and (**b**) succeeded, and (**c**) NB and (**d**) 2PR mechanical joining processes.

As can be seen in Figure 9a, for samples I+II, an air gap appeared in the bonding zone. Whereas, joining I+I samples by ICJ was not completely accomplished, as highlighted in Figure 10, where only a small area of the joining zone combined. Inadequate consolidation pressure caused insufficient bonding of type II+II samples, as shown in Figure 10a. A small zone connected, and the rest of the material remained unjointed. The same defects and gaps were also found during the induction joining of composite structures by the authors of [9].

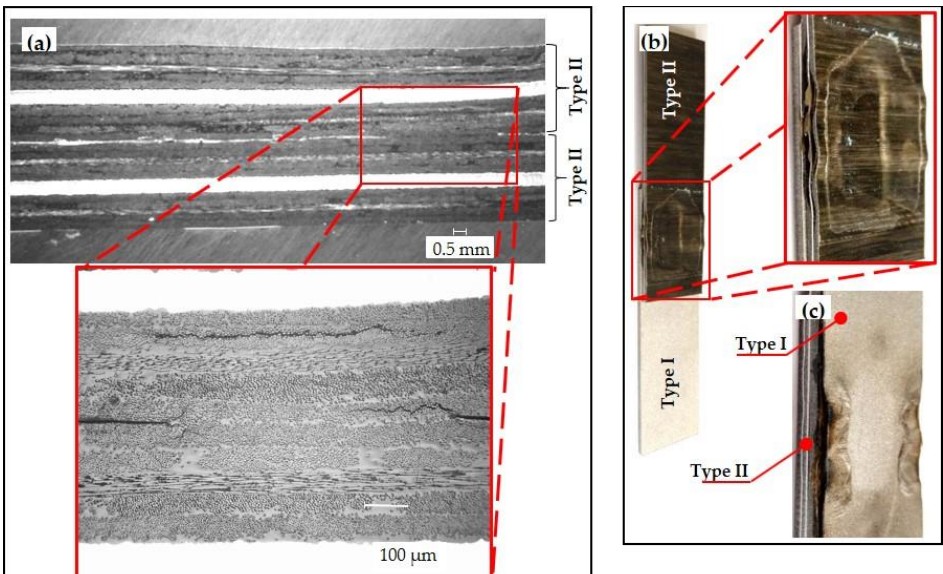

**Figure 10.** (**a**) Microscopic cross-section of ICJ type II+II samples with consolidation pressure of 2 bar, (**b**) sample I+II, 3 bar, with 5 kW induction power, and (**c**) sample I+II, same parameters as in (**b**), Type I is in the upper side.

The initial experiments used for the ICJ process Type I+II samples led to improving the optimum consolidation pressure and induction power to 3 bar and 4.5 kW, respectively. Figure 10b pinpoints the delamination and wavy shaped fiber layers for the enlarged view of ICJ Type I+II. Applying 4 bar consolidation pressure distorted the laminated surface, especially using induction power higher than 4.5 kW. Moreover, the higher pressure and induction power applied to the sample I+II resulted in melting the aluminum sheet of the Type I laminate, as clearly highlighted in Figure 10c.

The applied consolidation pressure with ideal induction power in addition to the optimum heating time in the ICJ resulted in successfully bonded samples, as shown in Figure 11. The enlarged view of Figure 11a shows that the start of the bonding zone appeared on the left side of the sample cross-sectional view after the mixing was started between the two laminates. The PA6 substrates were reconsolidated and the joint was consolidated. Moreover, a perfect consolidation of the PA6 matrix between the two laminates appeared clearly in Figure 11b in the enlarged cross-sectional view. The absence of air gaps and homogenous heating resulted in an optimum joining zone. The optimum consolidation pressure was investigated by the authors of [8] to improve the joined parameters correctly.

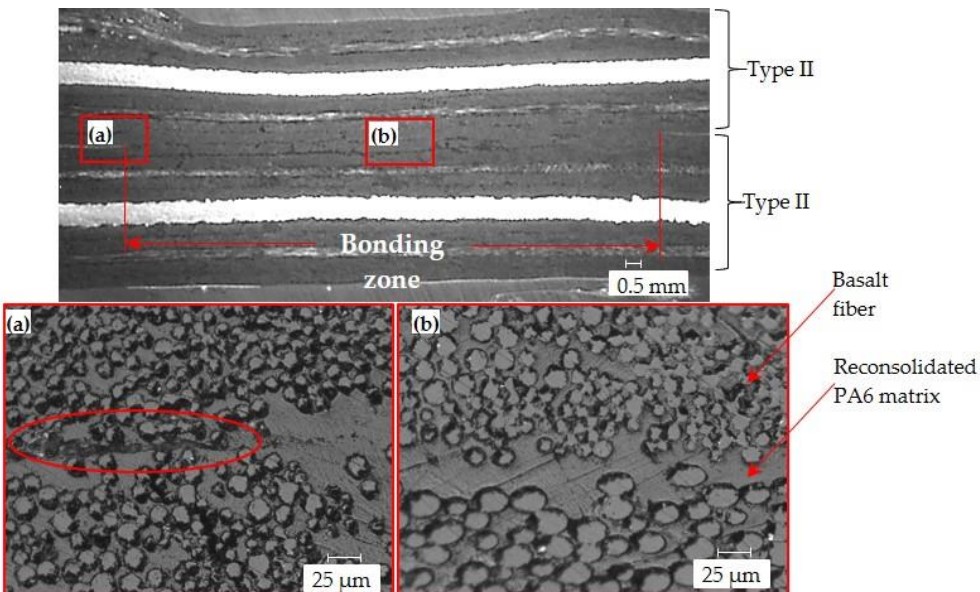

**Figure 11.** Joined sample of II+II showing the bonding zone length. (**a**) start of the bonding zone and (**b**) consolidation of the PA6 matrix between the two laminates.

### 5.3. Tensile Shear Test

The average mean values of the tensile shear tests for each joining process were calculated. The 2PR joining process showed the lowest tensile force, as presented in Figure 12, compared with that of the other two joining processes. Medium displacement values of the tested samples were between 3 to 4 mm, but the highest force was shown in I+II specimens. In addition, the appearance of the ruptured samples after the tensile shear tests is shown in Figure 12. The rivet was ruptured for all the sample types. Moreover, the BFRPL were not influenced by the broken rivet and the drilled holes had no delamination appearance.

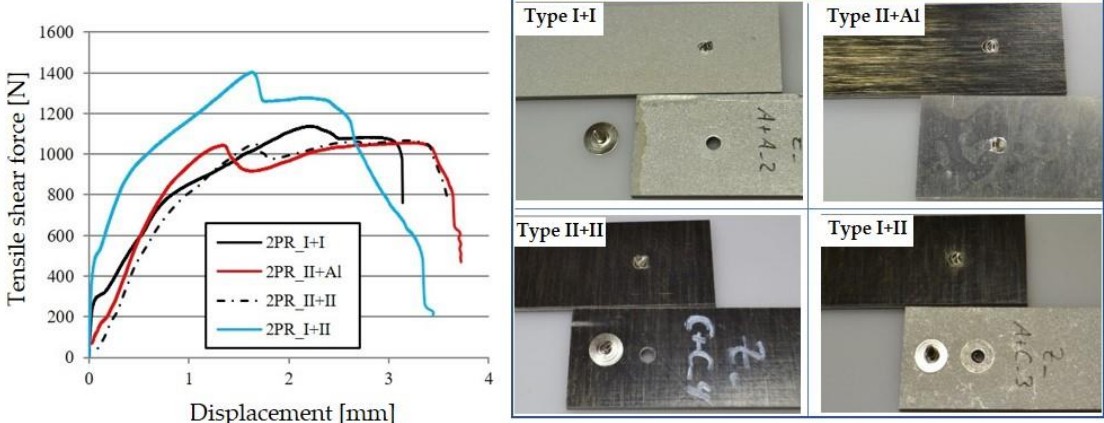

**Figure 12.** Tensile shear force and displacement curve with fractured joints after tensile shear test of two-part hollow rivet joining process.

The maximum displacement in the tensile shear test for the ICJ samples was found in sample II+II, as indicated in Figure 13. Furthermore, the II+SAl samples showed the greatest tensile shear force in the bonded zone before fracture for the ICJ process. The greatest force was found to be 3.547 kN. In contrast, the previous findings investigated by the authors of [9] only refer to 13 N in the tensile shear test, and the heating and cooling time for the samples was 160 s. The sandblasted aluminum sample II+SAl shows the highest tensile shear force due to the melted PA6 of BFRPL Type II inside the rough sandblasted aluminum surface.

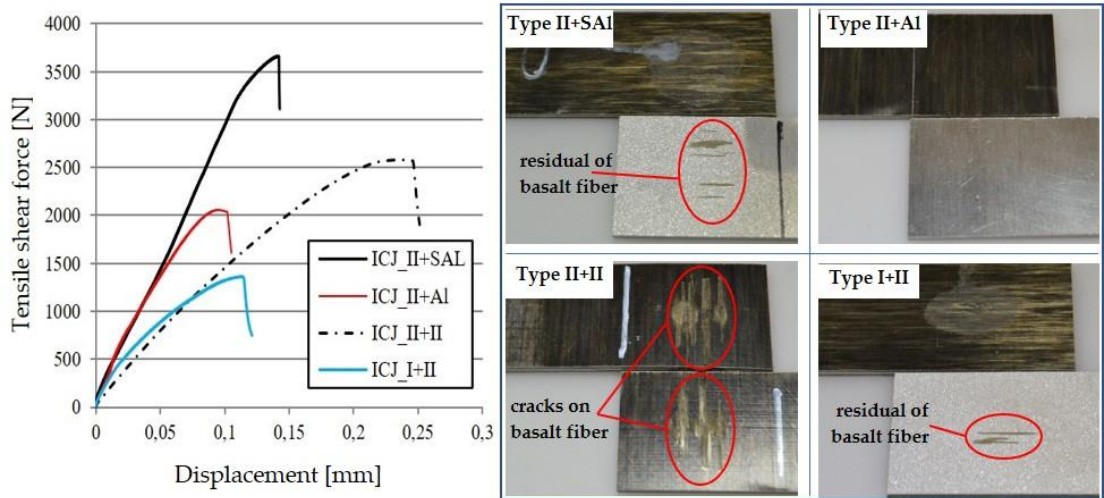

**Figure 13.** Tensile shear force and displacement curve with fractured joints after the tensile shear test of the ICJ process samples.

The partial residual of the PA6 matrix was clearly apparent in the fractured joining after the tensile shear tests, especially in the sample II+SAl. Moreover, cracks appeared on the surface of the combination of II+II, as can be seen in Figure 13. It is worth noting that the orientation of basalt fibers, either at 0° or 90° to the direction of the tensile shear test, did not affect the resulting force. The joined I+II samples show that a much lower residual amount of basalt fibers remained on the type I surface after testing compared with that of the II+SAl samples. The heat transfer rate for samples II+SAl with 2.4 mm aluminum thickness was 0.8 K/s, which was significantly less than the sample I+II with 0.3 mm aluminum thickness, where it was 16 K/s. Due to the thin aluminum sheet, the

temperature in the heated zone was not enough to allow sufficient adhesion between the joining partners.

In a comparison of the three joining methods, the samples of the NB process withstood the highest tensile shear forces, as shown in Figure 14. The NB samples can withstand more tensile shear forces; this can be explained by the solid bolt, whereas the 2PR process has only hollow rivets to connect the joining partners. In contrast, the bond strength of the ICJ largely depends on the adhesion of the polymer layer; this is consistent with previous results in [7,8]. Furthermore, the laminated structure was destroyed, and delamination occurred after the tensile shear test, especially for the Type II laminate.

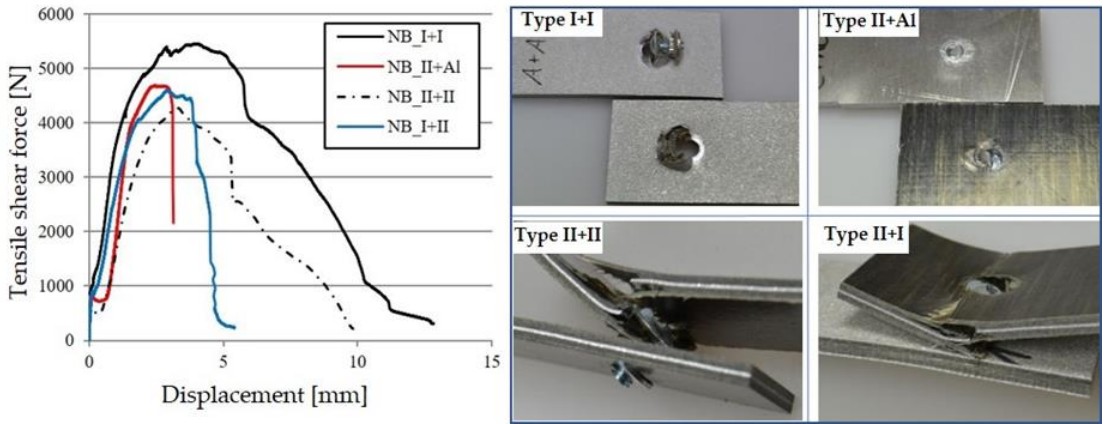

**Figure 14.** Tensile shear force and displacement curve with fractured joints after the tensile shear test of the nut and bolt joining process.

As a result, higher tensile shear forces can be achieved with the NB method than with the ICJ process. However, this is put into perspective by the considerable time required for preparation with drilling and subsequent fixing with a screw and nut. In contrast, the ICJ process requires only 12 s to join the laminate composite.

The premature failure of the base material is a likely mode of failure. Using the ICJ process, the fibers in the laminate structure were not affected by the joining process and the strength of the base material remained unaffected. This is an advantage of the ICJ process compared to the NB and 2PR joining methods. By analyzing the fractured surfaces of the tested parts joined by ICJ, it can be observed that only a fraction of the overlapping surface effectively formed a joint. The ICJ joining process can, therefore, be optimized to maximize the effective joining area, which is likely to increase the joint strength significantly.

## 6. Conclusions

Conditionally joinable BFRPL were successfully joined using the ICJ process. The results were compared with those of two other mechanical joining processes: bolt and nut and two-part hollow rivet. Tensile shear tests were carried out to determine the bearing strength of the specimens produced by the different joining methods. The ICJ process was able to achieve acceptable joint strength in a short time compared to the strength of the mechanical joints.

In addition, ICJ can easily be used to prepare laminates for joining them with other materials. The most effective parameters in the ICJ process were induction power, frequency, surface treatment, and heating time. Furthermore, the investigated ICJ process is easy to apply without additional fillers or adhesion materials. The investigated ICJ is still new and needs much more development through the testing of new parameters with preliminary surface treatments, chemical or mechanical, before joining, for either contact surface of joined partners and investigating other bonding polymer types.

**Author Contributions:** Conceptualization, A.A.-O., and J.K.; methodology, A.A.-O.; software, J.K.; validation, A.A.-O., J.K.; writing—original draft preparation, A.A.-O.; writing—review and editing, A.A.-O., V.K. and J.K.; supervision, V.K.; project administration, A.A.-O.; funding acquisition, V.K. All authors have read and agreed to the published version of the manuscript.

**Funding:** The publication of this article was funded by Chemnitz University of Technology. This research was funded by the Central Innovation Program for Medium-Sized Companies (ZIM) of the Federal Ministry for Economic Affairs and Energy.

**Informed Consent Statement:** Informed consent was obtained from all subjects involved in the study.

**Acknowledgments:** The financial support for the BasaltFaser—BasaOrth—(16KN021679) by the Central Innovation Program for Medium-Sized Companies (ZIM) of the Federal Ministry for Economic Affairs and Energy is gratefully acknowledged. The authors would like to thank the Cetex Institute for Textile and Processing Machines GmbH for manufacturing the BFRPL.

**Conflicts of Interest:** All the authors declare no conflict of interest.

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
