# Peer review of "Hybrid Joining by Induction Heating of Basalt Fiber Reinforced Thermoplastic Laminates"

_jcs, doi:10.3390/jcs5010010_

Round 1

Reviewer 1 Report

In this paper, the authors compared three different joining methods with an emphasis on the inductive contact joining. There are some shortcomings that must be addressed before it can be accepted for publication.

  1. The literature review in the introduction section is not sufficient as required by the Journal peer-review requirement, the authors need to expand the introduction of joining techniques other than just conventional mechanical fastening using extruded pins. Please revise as follows:

Mechanical joining methods were needed to join the polymer composites with metal sheets, like joining by extruded pins, as introduced by [3]. Other mechanical joining methods such as flow drill screw (FDS) and self-piercing riveting (SPR) have limitations when the materials to be joined have a high strength but low ductility. Novel fasteners/bolts using metal-ceramic composites have also been developed [Composite Structures 134 (2015) 680-688], which have high mechanical strength and low thermal expansion as the advantageous individual material properties such as the mechanical strength of metals and the heat and corrosion resistance of ceramics can be combined. Mechanical fastening utilising these novel fasteners/bolts could potentially be applied in aerospace [Composites Part B 82 (2015) 13-22, Ceramics International 42 (2016) 1416-1424], mechanical [Ceramics International 41 (2015) 8142-8148], and automotive applications.

  1. In Section 3, the inductive contact joining (ICJ) should be introduced in Section 3.1 before the other two joining methods. Although the ICJ is not a novel method firstly proposed in this paper, it has been published before in [18][19], the principle should at least be introduced here to make the readers understand it more easily without referring to other references. Currently, the ICJ process principle shown in Fig. 2 is far from being enough. More details should be given. For example, how is the cooling being operated? Where is the polyamide matrix and how is it being heated?
  2. The font size of axes in Fig. 7 is too small.
  3. In this paper, the optimal conditions of ICJ were defined as bonding without obvious air gaps. The mechanical tests reveal that only a fraction of the overlapping surface effectively formed a joint. Therefore, bonding without obvious air gap is not enough to form an optimal joining, what are the effects of process conditions on the effective bonding areas?
  4. Why is polyamide PA6 chosen as the melting matrix? Have the authors thought about other materials which might lead to a better performance?

Author Response

General reply to the reviewers for the manuscript

Hybrid joining by induction heating basalt fiber reinforced thermoplastic laminates

General reply to the first reviewer

We are very grateful to the first reviewer whose comments allow us to better clarify some important features of our approach. Below we reply to the first reviewer and we discuss the changes (highlighted in the manuscript in red colour) introduced to improve the manuscript.

  • We would like to thank the first reviewer for his helpful review as requested the introduction section was changed and additional reference related to join metal ceramic was added as can be seen in page 1.
  • It was modified as can be seen in page 3 in lines 119 till 123 and additional figure was introduced as illustrated in figure 2 (b).
  • We would like to thank the first reviewer for this notice the figure was modified as can be seen in page 8 for figure 7.
  • This was modified as can be seen in page 6 lines 171 till 175.
  • This is a very specific question and we would like to say that there are naturally other alternatives for the polymer AP6 to be applied in the experiments but this research was included to enhance and study specifically this type of laminated structure with another research institutes but certainly in the future we will take your advice in consideration. Therefore, we added this line to the conclusion at page 13 line 334.

Reviewer 2 Report

The manuscript titled "Hybrid joining by induction heating basalt fiber reinforced thermoplastic laminates" reports the inductive contact joining (ICJ) of basalt fiber reinforced polymer laminates. In addition to that, the authors compared the ICJ to two mechanical joining processes, nut and bolt and two-piece hollow riveting. The manuscript is well and clearly written. I recommend publishing the manuscript in the Journal of Composites Science.

Author Response

We want to thank the second reviewer for his comments on our manuscript

Round 2

Reviewer 1 Report

I have no further comments.